# Straw Stocks as a Source of Renewable Energy. A Case Study of a District in Poland

**Renata Marks-Bielska [1,*], Stanisław Bielski [2], Anastasija Novikova [3] and Kęstutis Romaneckas [4]**

1   Department of Economic and Regional Policy, Faculty of Economic Science, University of Warmia and Mazury in Olsztyn, Oczapowskiego 4, 10-719 Olsztyn, Poland

2   Department of Agrotechnology, Agricultural Production Management and Agribusiness, Faculty of Environmental Development and Agriculture, University of Warmia and Mazury in Olsztyn, 10-719 Olsztyn, Poland

3   Research Institute for Bioeconomy, Vytautas Magnus University, K. Donelaičio str. 58, 44248 Kaunas, Lithuania

4   Institute of Agroecosystems and Soil Sciences, Faculty of Agronomy, Agriculture Academy, Vytautas Magnus University, K. Donelaičio str. 58, 44248 Kaunas, Lithuania

*   Correspondence: renatam@uwm.edu.pl

**Abstract:** Biomass is playing an increasingly important role as a source of renewable energy. The aim of this study has been to identify the potential applicability of straw from agricultural crops to generate energy within the district of Braniewo, in the province Warmia and Mazury, Poland. The study covered the years 2015 to 2017. Based on statistical data, and using appropriate equations and norms, the structure of crop production and the number of livestock in the mentioned district were analysed; the potential production volume of straw was estimated, from which the amount needed for animal production (feed and bedding) was deducted, while the organic substance balance in soil was calculated. An annual average amount of straw remaining to be used for energy purposes in the district of Braniewo is about 41,531 t of straw, equivalent to about 60,222 GJ of energy (24,088 t of coal). In addition to the above analyses, a survey was conducted among local farmers, which showed their opinions about barriers to and opportunities for growing crops for energy purposes and using renewable energy resources. The survey results justify the claim that there is certain potential among farmers in the district of Braniewo to grow crops for energy purposes.

**Keywords:** straw; agriculture residues; biomass; bioenergy; Poland

## 1. Introduction

The growth of an economy of any size needs a constant supply of energy, the demand for which is constantly growing at an increasingly rapid rate. Maintaining the country's energy balance entails sustainable adjustment of the supply, both currently and in the long-term future, to the predicted demand for energy and fuels. The energy balance should take into account economic and ecological aspects, as well as possibilities of shaping the demand for energy without restraining the needs of consumers to be supplied energy. One of the priorities in the growth of energy generation in the near future is the development of renewable energy sources (RES). The share of RES in the fuel and energy balance contributes to the improved efficiency of energy consumption, helps to save fossil energy resources, and creates an opportunity to improve the quality of the natural environment, mostly by reducing the emission of pollutants and limiting quantities of waste. The predicted growth in the demand for primary energy in Poland to the year 2030 equals ca. 27% relative to the year 2010. The share of renewable energy in total energy consumption is expected to increase from around 8.8% in 2015 to 12% in 2020 and 12.4% in 2030 [1].

In line with the assumptions put forth in Poland's Energy Policy until the year 2030 [2] and the National Action Plan for Renewable Energy [3], the basic source of renewable energy in the province of Warmia and Mazury is to be biomass. Globally, biomass is the third largest natural source of energy. According to the definition proposed by the European Union, biomass consists of biodegradable products and their fractions, waste and residues from agricultural production (including plant and animal substances), forestry and related branches of economy, and biodegradable fractions of industrial and urban waste [4].

Biomass is studied and developed all over the world as a potential low-emission source of renewable energy. There are many strategies concerning renewable energy generated from available stocks of biomass [5–7]. However, global bioenergy markets may have an impact on resources of local food, feeds, and raw products for the food processing industry [8]. Agricultural biomass, as one of the major renewable energy resources in the world, is considered to be the source of greatest energy generation potential. With respect to the degree of processing, biomass can be divided into primary (annual and perennial energy crops—surplus green biomass from grasslands not used for animal feeding) and secondary biomass (waste—by-products from agricultural production food processing, liquid and solid animal faeces, organic residues from the agricultural and food industries; e.g., glycerin, spent distillery grains, slaughter waste, and dairy waste). The important role of agricultural biomass in the development of bioenergy generation is indisputable, although there are certain discrepancies in estimates of its potential [9–11].

Biomass is a typical local fuel and should be utilized at a site of production by individual customers as the main fuel for distributed cogeneration power plants producing electric power and thermal energy. Using biomass for energy purposes depends on a number of factors, including economic incentives [12], or available technologies. Biomass production depends to the greatest extent on the prices of energy raw materials and the assurance of long-term sales. Guidelines for agricultural biomass management in Poland are contained in the Strategy for Sustainable Development of the Countryside, Agriculture and Fisheries for 2012–2020 [13]. An exhaustive description of this issue is presented in Specific Aim 5: Protection of the environment and adaptation to climate change in rural areas, in Priority 5.3; adaptation of agriculture and fisheries to climate change, and their participation in the mitigation of climate change, in Priority 5.5; and raising the use of renewable energy resources in rural areas.

The development of renewable energy sources (RES), as well as their rational utilization, is a universal measure of responsible environmental policy. Using energy from renewable sources, i.e., water, wind, solar radiation, geothermal energy, or biomass, can bring about measurable ecological and energy-related outcomes, thus contributing to the dissemination of the notion of sustainable growth [1]. This concept rests on three pillars: (1) Ecological (the environment is used in such a way as to preserve natural resources without decreasing the quality of life for society), (2) economic (gaining the desirable revenues and economic growth), and (3) social (ensuring adequate quality of life while respecting cultural identity) [14,15]. As the idea of sustainable development is being disseminated, it spreads over new areas of economic activity, including energy generation. Under the framework of sustainable development, three principles are mentioned in regard to energy sources, all corresponding well to the underlying assumption of the above theory: (1) Resources of sustainable energy will not be depleted due to their further utilization; (2) the use of these energy sources does not cause any larger emission of pollutants; and (3) the use of these energy sources does not contribute to greater social injustice, nor does it increase the risk to human health [16]. In compliance with the report of the United Nations, called Our Common Future [17], it is justifiable to claim that as of now we do not know of any source of energy that would be completely sustainable.

The growing use of renewable energy sources depends on the rate of technological progress, on the pressure of various groups lobbying to reduce emission of greenhouse gases, and above, all on the possibilities of creating an integrated system of biomass production, converting biomass into energy and possessing an energy transmission grid. The efficient use of biomass for energy generation could

be supported by the formation of local, distributed energy centres, situated in rural areas. Building a local system for biomass conversion to energy is a solution considered to be, energy-wise—efficient, fully ecological, and stimulating to rural areas (new jobs, more efficient land use, and capital turnover in a local system) [10].

Studies designed to recognise the conditions which have influence on the production of biomass for energy purposes in the Province of Warmia and Mazury have demonstrated that biomass producers are mostly concentrated in the western part of the region and in a belt stretching through the north-eastern part of the province [10]. An important consideration in the establishment of agricultural biomass plantations is the localisation relative to potential biomass delivery points. The key factor influencing the profitability of biomass production is the transportation cost, which depends not only on the distance over which energy feedstock is carried, but also on the type of raw material and degree to which it has been processed. Ensuring satisfactory revenue is possible when the localisation of a plantation is optimal, and the type of feedstock produced satisfies the needs of clients [18].

From the point of view of entrepreneurs involved in the purchase and conversion of biomass to energy generation purposes, the north-western part of the Province of Warmia and Mazury stands out against the background of the whole region. The localisation of relatively large enterprises which are potential buyers of biomass feedstock is conducive to the concentration of field biomass production for energy purposes. In the central part of this region, there are single enterprises of this kind, whereas in the northern and south-eastern belts there are few biomass buyers and producers. A relationship has been noted between the localisation of biomass conversion plants and farms engaged in the production of biomass for energy purposes in the municipalities and districts where the IAPAQ is on a moderate level [10]. The Index of Agricultural Production Area Quality (IAPAQ) is an aggregate index based on the assessment of indicators (diagnostic features), such as: Soil quality, climate, land relief, and water balance. These factors are described by ordered parameters assigned weights that reflect the relative scale of importance of each factor for soil fertility. By tallying the weighted factor scores, a synthetic numerical indicator is obtained to characterise the quality of an agricultural production area within the theoretical range of 20 to 122 points, which mirrors the potential natural productivity of a given area.

The Local Government Act [19] specifies that heat supply is a task delegated to municipalities. The Energy Law [20] (article 18.1) states that a municipality is a territorial unit responsible for planning and organising a system of heat supply in its territory. A correctly developed and implemented project of a plan to supplying heat, electric power, and gas fuels is one of the essential conditions for ensuring sustainable development of urban and rural areas. The use of renewable energy sources (RES) inscribes itself in these plans, being an integral component of sustainable development. In compliance with the Energy Law, the developed projects should include a feasibility study of employing local energy resources.

Agriculture and energy policy are two closely interrelated elements. Energy crops in the future may become a strategic direction in agricultural production, and thereby contribute to an increase in the share of biofuels, greater energy supply, and the achievement of the current energy policy's targets [9]. In Poland, there is a large untapped potential of agricultural production, which can be taken advantage of to generate energy. It would be recommended to take measures, so as to convert all biomass unused so far, representing surplus production, especially if is low-quality, to energy.

Agricultural biomass is comprised of straw (ripe or dried stems of cereals). Dried stems of leguminous plants, flax, or oilseed rape are also referred to as straw. Straw is a residual material left on farms after a plant production cycle. Utilisation of straw by the power generation industry seems to be an optimal solution, as it brings about versatile benefits. Moreover, straw is the second most important biofuel in Poland, after wood. In recent years, as new technologies have developed, using straw for energy purposes has become more widespread and economically viable. It allows us to save energy and to reduce emission of airborne pollutants [21,22]. Straw as feedstock for power generation may become very important in rural areas, where it appears in amounts exceeding the potential utilisation

for agricultural purposes. Other than farms, straw can be utilised by low and medium-sized power plants, entire housing estates, school buildings, municipal office buildings, hotels, etc.

A review of the consequences of removing agricultural by-products in the context of sustainable development has revealed the environmental factors which can inhibit this process [23]. Removal of agricultural residues should be carried out with great care. Such leftovers play a number of vital roles in an agronomic ecosystem, including their direct and indirect impact on physical, chemical, and biological processes in soil [23–25].

In other countries, the growing interest in straw as a source of energy has encouraged making more detailed assessments of its availability for bioenergy production [22,26–32]. For instance, it has been estimated that the total UK crop residues equate 20.4 Mt dry matter production, of which 8.37 Mt were collectable and 4.2 Mt were available [33].

Another trend observed in recent years is to use straw as second-generation biofuels (fuels produced from agricultural non-food produce). Second-generation bioethanol is a liquid biofuel used for transportation purposes, in line with the sustainable development assumptions, as both the generation of ethanol and its combustion in engines contributes to diminishing the emission of $CO_2$ to the atmosphere [34–37]. As well as being used to generate heat energy, straw is converted to make biogas [38–40].

Until the present, it has been impossible to make liquid biofuels from non-food cellulose plant feedstocks on the industrial scale due to economic and technological barriers [41,42]. Researchers all over the world pay more attention to the development of a technology for production of cellulose-based ethanol using regional stocks of biomass, which would decrease the cost of raw materials and improve the efficiency of such technology, leaving the smallest possible trace in the natural environment [35,43–57].

Straw is a very popular renewable energy source. It is extremely difficult to make a precise straw balance on a macro-scale (a province or the whole country) due to certain clusters of production areas within such large territorial units [58,59]. More reliable results can be achieved for smaller units of the country's administrative division, such as a municipality or a district, or for large farms [60,61].

The directions of actions implicated by the regional policy implemented in Poland as being able to make the largest contribution to increasing the share of energy from renewable sources in the total energy balance of the Province of Warmia and Mazury are described in the Concept of the Development of RES until 2020 for this province [62]. The document emphasises that the achievement of the 15% share of renewable energy in the final energy consumption in 2020, as required in the energy and climate package, puts an obligation on local governments to support the existing and encourage the construction of new facilities using new sources of energy, the aim of which will be to develop the region and make it independent from external supplies of power.

The main goal of this study has been to determine the potential of using straw from energy crops for energy purposes in the district of Braniewo. The structure of crops and the livestock in the municipalities within the district were analysed, and the potential amount of straw available for energy purposes was estimated. The research covered municipalities in the district of Braniewo, the Province of Warmia and Mazury, as this is an agricultural region where agricultural biomass can be collected. All analyses were supplemented with results of a survey study, whose aim was to find out farmers' opinions about barriers to and opportunities for growing energy crops on their farms, and about using renewable energy sources.

## 2. Materials and Methods

In order to calculate potential stocks of straw biomass in the municipalities within the district of Braniewo, data concerning the structure of crops and numbers of livestock were collected from each municipality. The structure of crops was determined according to data made available by the Warmia and Mazury Agricultural Advisory Centre in Olsztyn [63]. Our analysis of the structure of cropped fields is based on information from three years (2015, 2016, and 2017). The information about

the number of livestock comes from the databank of the Olsztyn Regional Branch of the Agency for Restructuring and Modernisation of Agriculture.

The following formula served to calculate yields of straw from particular crops:

$$P = \sum_{i=1}^{n} a \cdot y \cdot wzs \tag{1}$$

where:

$P$—production of straw of basic cerelas and oilseed rape,
$a$—area cropped with the $i$-th species of a given crop [ha],
$y$—grain yield of the $i$-th species of the crop [t·ha$^{-1}$], and
$wzs$—grain to straw ratio.

In order to estimate properly the potential amount of straw that could be used for energy purposes, the total straw yields should be made less by the amount of straw consumed by farming. First of all, straw needs to cover the demand of animal production (bedding and feed) and, if necessary, needed to maintain a sustainable balance of organic matter in soil. The following formula supported our calculations:

$$N = P - (Zs + Zp + Zn) \tag{2}$$

where:

$N$—surplus of straw for alternative (energy generation) use,
$P$—production of straw from basic cereals and oilseed rape,
$Zs$—demand for straw for bedding,
$Zp$—demand for straw for animal feed, and
$Zn$—demand for straw to be ploughed into soil.

The demand for straw to be used in animal production (bedding and feed) was calculated according to the following equations:

$$Z_s = \sum_{i=1}^{n} qi \cdot si \tag{3}$$

$$Z_p = \sum_{i=1}^{n} qi \cdot pi \tag{4}$$

where:

$Z_s$—demand for bedding straw,
$Z_p$—demand for feed straw,
$qi$—livestock of the $i$-th species and purpose group,
$si$—standard demand for bedding straw of the $i$-th species and purpose group, and
$pi$—standard demand for feed straw of the $i$-th species and purpose group.

When calculating the demand for straw needed to be ploughed into soil, it was necessary to take into account the structure of crops, quality of soils, and balance of organic substance. An increase or decrease in the soil content of organic matter can be calculated with the help of coefficients defining its reproduction or degradation (Table 1) [64]. The adopted reproduction or degradation coefficient was the one assigned to medium rich soils, as these are prevalent in the district of Braniewo. With the known area cropped with particular groups of plants and the amount of produced manure, which was computed from the number of livestock and relevant standards, the balance of organic substance was calculated from:

$$S = \sum_{i=1}^{n} ri \cdot wri + \sum_{i=1}^{n} qi \cdot oi \tag{5}$$

where:

*S*—balance of organic substance,
*ri*—area cropped with a given group of plants,
*wri*—coefficient of reproduction or degradation of organic substance for the given group of plants,
*qi*—number of livestock in live heads, according to species and age groups, and
*oi*—standards of manure production in t per year according to species.

**Table 1.** Coefficient of reproduction or degradation of organic substance in soil depending on the crop species.

| Species | Coefficient |
|---------|-------------|
| Basic cereals | −0.53 |
| Oilseed rape | −0.53 |
| Potato | −1.40 |
| Maize | −1.15 |
| Legumes | 0.35 |

Source: The authors, based on Pruszek [64].

For our calculations, we considered the numbers of livestock in the municipalities of the district of Braniewo and the annual standards for particular species and purpose types included in Abbreviated Agricultural Production Standards [65].

As the balance of organic substance in soil was found to be negative, it was necessary to plough into soil, some amount of straw in order to maintain a balanced content of organic matter in soil (assuming that 1 tonne of dry matter of manure corresponds to 1.54 tonne of straw). To calculate the balance of organic substance, the following formula was employed [66]:

$$Zn = ws/o \cdot s \tag{6}$$

where:

*Zn*—demand for straw to be ploughed into soil,
*ws/o*—the 1.54 Mg straw coefficient equating 1 Mg of manure dry matter, and
*s*—balance of organic substance.

The average market price of straw in transactions done between agricultural producers, as of the second quarter of 2019 in the Province of Warmia and Mazury, was included in our calculations.

The analyses were supplemented with the results of a survey addressed in 2018 to farmers in the district of Braniewo. Because there are no statistics showing the number of agricultural biomass producers, to obtain such information we conducted uncategorised, direct interviews with employees of the Warmia and Mazury Agricultural Advisory Centre, who helped to gather the information needed for this study from agricultural advisors working in the territory of the district of Braniewo, heads of local government in villages, and with an entrepreneur who purchased straw in the analysed district and converted it into energy. Plantations of energy crops (willow, miscanthus, Virginia fanpetals) were identified, and their owners were reached. Potential producers of biomass for energy purposes were also searched out among farmers engaged in organic farming and in non-farming economic activities. Our efforts led to the identification of 35 farmers who answered the questionnaire.

## 3. Results and Discussion

*3.1. Analysis of Local Agricultural Biomass Resources in the District of Braniewo*

There are many factors involved in straw production. Of these, the most important ones are: Field areas cropped with plants, yields, species of crops, fertilisation, and cultivars—especially ones

with rigid or short stems, where the straw to grain yield ratio is lower. Average yields of crops grown in the district of Braniewo were calculated on the basis of data gathered by the Agricultural Advisory Centre in Olsztyn. To calculate the yields of straw, the coefficients of main to secondary yields were used (Tables 2–4).

The following straw was included in our analysis: Straw from cereals grown for grain (wheat, rye, barley, oat, and triticale), and straw of oilseed rape. The straw to grain yield ratios for these crops were taken after: Denisiuk [67] and Grzybek et al. [68].

**Table 2.** Conversion coefficient of side yield (straw) to main yield.

| Species | Coefficient |
|---|---|
| Winter wheat | 0.46 |
| Spring wheat | 0.46 |
| Winter barley | 0.70 |
| Sporing barley | 0.78 |
| Rye | 1.45 |
| Triticale | 1.13 |
| Oat | 1.05 |
| Mixed cereals | 1.10 |
| Winter oilseed rape | 1.00 |
| Spring oilseed rape | 1.00 |

Source: The authors, based on Denisiuk [67] and Grzybek et al. [68].

**Table 3.** Area cropped and yields of crops producing straw, as well as potential straw yield in the district of Braniewo (means from the three years, 2015 to 2017).

| Species | Cropped Area [ha] | Yield [t·ha$^{-1}$] | Potential Straw Production [t·year$^{-1}$] |
|---|---|---|---|
| municipality Braniewo | | | |
| Winter wheat | 2450 | 4.6 | 4867 |
| Spring wheat | 423 | 4.0 | 730 |
| Winter barley | 39 | 3.8 | 105 |
| Spring barley | 373 | 3.6 | 983 |
| Rye | 242 | 3.9 | 1402 |
| Winter triticale | 308 | 4.3 | 1440 |
| Spring triticale | 52 | 3.5 | 219 |
| Oat | 610 | 3.7 | 2352 |
| Mixed cereals | 270 | 3.6 | 990 |
| Winter oilseed rape | 1267 | 3.1 | 3927 |
| Spring oilseed rape | 240 | 1.7 | 416 |
| Total in Braniewo municipality | 6274 | - | 17,430 |
| municipality Frombork | | | |
| Winter wheat | 847 | 4.2 | 2080 |
| Spring wheat | 295 | 3.6 | 74 |
| Winter barley | 0 | 0.0 | 0.0 |
| Spring barley | 119 | 3.9 | 181 |
| Rye | 88 | 3.6 | 388 |
| Winter triticale | 125 | 3.8 | 238 |
| Spring triticale | 20 | 3.3 | 129 |
| Oat | 223 | 3.6 | 674 |
| Mixed cereals | 152 | 3.4 | 481 |
| Winter oilseed rape | 172 | 2.5 | 423 |
| Spring oilseed rape | 41 | 1.6 | 63 |
| Total in Frombork municipality | 2082 | - | 4732 |

**Table 3.** *Cont.*

| Species | Cropped Area [ha] | Yield [t·ha$^{-1}$] | Potential Straw Production [t·year$^{-1}$] |
|---|---|---|---|
| | municipality Lelkowo | | |
| Winter wheat | 1565 | 4.1 | 3386 |
| Spring wheat | 278 | 3.4 | 213 |
| Winter barley | 0 | 0.0 | 0.0 |
| Spring barley | 158 | 3.3 | 229 |
| Rye | 393 | 3.4 | 1494 |
| Winter triticale | 313 | 4.0 | 1540 |
| Spring triticale | 47 | 3.2 | 163 |
| Oat | 620 | 3.7 | 2366 |
| Mixed cereals | 352 | 3.4 | 1133 |
| Winter oilseed rape | 403 | 2.8 | 1116 |
| Spring oilseed rape | 70 | 1.5 | 102 |
| Total in Lelkowo municipality | 4199 | - | 11,741 |
| | municipality Pieniężno | | |
| Winter wheat | 1792 | 4.4 | 4058 |
| Spring wheat | 730 | 3.7 | 996 |
| Winter barley | 45 | 3.5 | 107 |
| Spring barley | 507 | 3.8 | 1186 |
| Rye | 468 | 3.6 | 2327 |
| Winter triticale | 439 | 4.1 | 2632 |
| Spring triticale | 57 | 3.4 | 124 |
| Oat | 997 | 3.6 | 3483 |
| Mixed cereals | 603 | 3.5 | 2288 |
| Winter oilseed rape | 817 | 3.0 | 2450 |
| Spring oilseed rape | 223 | 2.0 | 454 |
| Total in Pieniężno municipality | 6677 | - | 20,104 |
| | municipality Płoskinia | | |
| Winter wheat | 2150 | 4.7 | 4937 |
| Spring wheat | 482 | 3.9 | 633 |
| Winter barley | 0 | 0.0 | 0.0 |
| Spring barley | 287 | 3.7 | 728 |
| Rye | 348 | 3.8 | 2694 |
| Winter triticale | 445 | 4.0 | 1990 |
| Spring triticale | 43 | 3.4 | 217 |
| Oat | 607 | 3.9 | 2842 |
| Mixed cereals | 200 | 3.6 | 785 |
| Winter oilseed rape | 800 | 3.1 | 2453 |
| Spring oilseed rape | 168 | 1.6 | 274 |
| Total in Płoskinia municipality | 5530 | - | 17,555 |
| | municipality Wilczęta | | |
| Winter wheat | 1578 | 4.7 | 3859 |
| Spring wheat | 625 | 3.9 | 907 |
| Winter barley | 25 | 3.6 | 75 |
| Spring barley | 362 | 3.6 | 918 |
| Rye | 67 | 3.6 | 209 |
| Winter triticale | 342 | 4.3 | 1692 |
| Spring triticale | 71 | 3.7 | 333 |
| Oat | 475 | 4.0 | 1604 |
| Mixed cereals | 403 | 3.5 | 1348 |
| Winter oilseed rape | 523 | 3.0 | 1586 |
| Spring oilseed rape | 118 | 1.8 | 212 |
| Total in Wilczęta municipality | 4589 | - | 12,743 |

Source: The authors, based on [63].

**Table 4.** Cropped area and yields of crops producing straw, and straw potential quantity in the district of Braniewo (means from the three years, 2015 to 2017).

| Species | Cropped Area [ha] | Yield [t·ha$^{-1}$] | Potential Straw Production [t·year$^{-1}$] |
|---|---|---|---|
| Winter wheat | 10,382 | 4.4 | 23,186 |
| Spring wheat | 2833 | 3.8 | 3553 |
| Winter barley | 109 | 1.8 | 287 |
| Spring barley | 1805 | 3.6 | 4225 |
| Rye | 1606 | 3.6 | 8513 |
| Winter triticale | 1972 | 4.1 | 9532 |
| Spring triticale | 291 | 3.4 | 1186 |
| Oat | 3531 | 3.7 | 13,320 |
| Mixed cereals | 1980 | 3.5 | 7025 |
| Winter oilseed rape | 3981 | 2.9 | 11,956 |
| Spring oilseed rape | 860 | 1.7 | 1521 |
| Total | 29,351 | - | 84,305 |

Source: The authors, based on [63,67,68].

In the calculations of the straw potential in the municipalities of the district of Braniewo, the following were considered: Area of farmland cropped with plants producing straw, volume of yields of crops, and ratios of main to side yields. From the above data, the amount of straw that could be obtained from the cropped farmland was calculated at 84,305 t.

The demand for bedding straw depends primarily on the number of livestock, and on the type of housing and animal housing facilities. Three types of animal housing are most popular: Shallow litter, deep litter, and litterless systems. Estimates suggest that in Poland about 80% of farm animals are kept on shallow litter, around 15%–20% on deep litter, and just 3%–5% in litterless facilities. For the sake of our analysis, it was assumed that 100% of animal housing facilities belonged to shallow litter ones. It was expected that a larger consumption of straw in deep litter sheds is set off by the savings achieved in litterless sheds [69]. In order to determine the demand for bedding and feed straw, annual standards defined for particular species and groups of farm animals were used. Table 5 presents data on the number of particular groups of farm animals in the analysed municipalities, as well as the annual demand for feeds and litter. Table 6 presents aggregate data for the district of Braniewo.

The final straw balance included the demand for straw needed to be ploughed in so far as to maintain the proper balance of organic substances in soil. The livestock kept in the district of Braniewo supplies 11,217 t of dry matter of manure. This is insufficient to cover the reduction of organic matter in soil (−18,083 t) (Tables 7 and 8). According to our calculations, the balance of organic substance in soil in the district of Braniewo is negative (−0.22 t d.m. of manure·ha$^{-1}$). The negative balance means that 10,573 t of straw must be ploughed into soil in order to maintain the proper balance of humus in soil. First of all, oilseed rape straw should be ploughed in because it is practically useless in animal rearing or for other purposes on a farm. Thus, it can be used as fertiliser or fuel. Moreover, it has a higher content of nitrogen than cereal straw. After ploughing in oilseed rape straw, it becomes unnecessary to add a supplementary dose of nitrogen; this straw decomposes in soil faster than cereal straw and contains two—to three—fold more sulphur than the latter; finally, it does not create a risk for the transmission of cereal fungal diseases.

The annual production of oilseed rape straw in the analysed district is 13,477 t, of which 2904 t a year can be used for energy purposes. In this analysis, because of environmental benefits, the surplus of oilseed rape will not be used for thermal energy generation. The demand for bedding straw in the analysed district equals 15,667 t, and 13,630 t of straw are needed for animal feeds. The above calculations show that 41,531 t of straw for energy purposes remain for alternative uses.

**Table 5.** Number of livestock in the district of Braniweo, and demand for straw for agricultural purposes.

| Species | Number of Animals [indiv.] | Demand for Feed [Mg] | Demand for Bedding [Mg] |
| --- | --- | --- | --- |
| municipality Braniewo | | | |
| Cattle | 3659 | 2561 | 2317 |
| Swine | 3926 | 0 | 1374 |
| Goats | 24 | 5 | 5 |
| Total in municipality Braniewo | | 2566 | 3696 |
| municipality Frombork | | | |
| Cattle | 962 | 674 | 609 |
| Swine | 148 | 0 | 52 |
| Goats | 19 | 4 | 4 |
| Total in municipality Frombork | | 678 | 665 |
| municipality Lelkowo | | | |
| Cattle | 2761 | 1933 | 1749 |
| Swine | 333 | 0 | 117 |
| Goats | 22 | 4 | 4 |
| Total in municipality Lelkowo | | 1 937 | 1 870 |
| municipality Pieniężno | | | |
| Cattle | 5724 | 4007 | 3625 |
| Swine | 1278 | 0 | 447 |
| Goats | 50 | 10 | 10 |
| Total in municipality Pieniężno | | 4017 | 4082 |
| municipality Płoskinia | | | |
| Cattle | 1964 | 1375 | 1244 |
| Swine | 1674 | 0 | 586 |
| Goats | 10 | 2 | 2 |
| Total in municipality Płoskinia | | 1377 | 1832 |
| municipality Wilczęta | | | |
| Cattle | 4362 | 3054 | 2763 |
| Swine | 2161 | 0 | 756 |
| Goats | 10 | 2 | 2 |
| Total in municipality Wilczęta | | 3056 | 3521 |

Source: The authors, based on the databank of the Olsztyn Regional Branch of the Agency for Restructuring and Modernisation of Agriculture, and [65].

**Table 6.** Number of livestock in the district of Braniewo, and demand for straw for agricultural purposes.

| Species | Number of Livestock [Indiv.] | Demand for Straw [Mg] | Demand for Bedding [Mg] |
| --- | --- | --- | --- |
| District of Braniewo | | | |
| Cattle | 19,433 | 13,603 | 12,308 |
| Swine | 9520 | 0 | 3332 |
| Goats | 136 | 27 | 27 |
| Total in District of Braniewo | | 13,630 | 15,667 |

Source: The authors, based on the databank of the Olsztyn Regional Branch of the Agency for Restructuring and Modernisation of Agriculture, and [65].

The market price of straw for energy purposes depends mainly on the calorific value, and in practice on the moisture content. Prices of straw as raw material vary. In our calculation, the average monetary value of 1 tonne of straw was assumed to be ca €32. However, it needs to be added that services required to handle straw (pressing, piling, and transport) in every case are arranged and paid for by the buyer. This means that the above price refers to straw left on a field after grain harvest. Technically speaking, the most appropriate content of water in straw should be below 15%. This is

very difficult to achieve and therefore it is acceptable that the moisture content in straw equals 20% or even 25%. Specific terms of contract are negotiable. The price of purchased straw is often proportional to the acreage of fields, and inversely proportional to the distance from fields to the buyer. We did not find in the district submitted to our analysis any business enterprises which would express the need to use straw for energy purposes (heat generating plants using biomass, or other economic entities; e.g., producers of substrates for mushroom growing, or producers of insulation materials).

**Table 7.** Balance of organic matter in soils of the district of Braniewo.

| Specification | Area Cropped [ha] | Balance of Organic Matter [t] |
|---|---|---|
| municipality Braniewo | | |
| Basic cereals | 4715 | −2499 |
| Oilseed rape | 1267 | −671 |
| Potato | 207 | −289 |
| Maize | 443 | −510 |
| Legumes | 84 | 29 |
| Total in municipality Braniewo | 6715 | −3940 |
| municipality Frombork | | |
| Basic cereals | 1850 | −980 |
| Oilseed rape | 172 | −91 |
| Potato | 207 | −289 |
| Maize | 19 | −22 |
| Legumes | 84 | 29 |
| Total in municipality Frombork | 2331 | −1353 |
| municipality Lelkowo | | |
| Basic cereals | 3679 | −1950 |
| Oilseed rape | 403 | −214 |
| Potato | 133 | −187 |
| Maize | 61 | −71 |
| Legumes | 73 | 26 |
| Total in municipality Lelkowo | 4350 | −2395 |
| municipality Pieniężno | | |
| Basic cereals | 5580 | −2957 |
| Oilseed rape | 817 | −433 |
| Potato | 128 | −179 |
| Maize | 605 | −696 |
| Legumes | 300 | 105 |
| Total in municipality Pieniężno | 7430 | −4160 |
| municipality Płosknia | | |
| Basic cereals | 4518 | −2395 |
| Oilseed rape | 800 | −424 |
| Potato | 30 | −42 |
| Maize | 336 | −386 |
| Legumes | 95 | 33 |
| Total in municipality Płoskinia | 5779 | −3214 |
| municipality Wilczęta | | |
| Basic cereals | 3877 | −2055 |
| Oilseed rape | 523 | −277 |
| Potato | 148 | −208 |
| Maize | 430 | −495 |
| Legumes | 41 | 14 |
| Total in municipality Wilczęta | 5020 | −3020 |

Source: The authors, based on [63,64].

**Table 8.** Balance of organic matter in soils of the district of Braniewo.

| Specification | Area Cropped [ha] | Balance of Organic Matter [t] |
|---|---|---|
| Basic cereals | 24,219 | −12,836 |
| Oilseed rape | 3981 | −2110 |
| Potato | 853 | −1194 |
| Maize | 1895 | −2179 |
| Legumes | 676 | 237 |
| Total in the district of Braniewo | 31,625 | −18,083 |

Source: The authors, based on [63,64].

### 3.2. Barriers to and Opportunities for Growing Crops for Energy Purposes on Farms, and Using Renewable Energy Resources, According to Farmers from the District of Braniewo

The changing socio-economic circumstances have resulted in the creation of a new function of farming, such as growing energy crops. Farmers and entrepreneurs engaged in this business gain higher profits, put cultivated or purchased crops to a better use, and help protect the environment. Only two persons engaged in energy crops have been identified in the district of Braniewo (5.7% of the surveyed farmers); one grew willow and Virginia fanpetals, and the other one cultivated miscanthus.

In the past, 5.7% of our respondents grew oilseed rape and cereals, while 20.0% intend to grow crops for energy purposes, including: Oilseed rape, cereals, willow, Virginia fanpetals, miscanthus, and Jerusalem artichokes. Furthermore, as many as 68.6% of the respondents have not grown and are not going to grow energy crops, of which 45.8% pointed to the lack of profitability, 37.5% admitted to foreseeing problems selling such crops, and the others (16.7%) provided other explanation; e.g., lack of stable prices, having no interest, or excessive labour consumption. The research showed that energy crop plantations in the district of Braniewo have not, are not, and most probably will not become a widespread farming practice.

Other questions in the survey concerned: A plan to construct an on-farm biogas plant, or intention to connect one's farm to a thermal energy grid supplied by an agricultural biomass plant. The vast majority of the respondents were not in the least interested in building an on-farm biogass plant (94.3%). The reasons were: High cost (35.0%), lack of feedstock (27.5%), low profitability (20.0%), negative attitude of the local community (15.0%), and having no such need (2.5%).

Only 2.9% of the respondents would like to build a biomass plant to produce and sell energy because of environmental considerations. For comparison, 42.9% would be interested in having their farm connected to a heating network supplied energy from biogas plants, mainly because it would be a less expensive and more ecological source of heat (71.4% and 28.6%, respectively). More than half the respondents (51.4%) were not interested in this solution, mostly because of predicted costs, having no such opportunity, or lacking conditions which would allow them to make such a connection.

One of the ways in which farmers or entrepreneurs can cut down costs of business is to produce their own biodiesel. Unfortunately, only 5.7% of the farmers from the district of Braniewo who participated in our study are going to produce biodiesel for own purposes, mostly because of lesser environmental pollution (66.7%) and a chance of producing less expensive fuel than conventional one (33.3%).

As many as 91.4% had no intention to produce biodiesel, mostly because of the lack of raw material (31.8%), complicated legal regulations (29.5%), low profitability of this type of production (20.5%) and the local community being against such investments (13.6%). Other arguments against the production of biodiesel on farms include the absence of coherent law concerning diesel oil, and the lack of right conditions on farm to carry out diesel production.

Another way of obtaining energy from renewable resources is by the combustion of biomass, such as straw. Very few of our respondents from the district of Braniewo, at the time of answering to our questionnaire, used straw for energy purposes on their farms (5.7%). The results showed that 17.1% of those questioned used straw for this purpose in the past but already stopped this practice, and only

2.9% intended to use straw for energy generation. The vast majority of the respondents (74.3%) did not use straw for energy purposes at all. It can be concluded that in total 91.4% (including the persons who have ceased using straw for energy generation) did not consider straw as an energy resource. The main arguments were: Lack of surplus straw (71.4%), low profitability (21.4%) and other reasons (7.1%).

In order to raise their revenues, farmers often sell surplus produce. Among the surveyed farmers from the district of Braniewo, only 8.6% currently sell straw for energy purposes, to make pellets and briquettes, or directly to heating plants.

Our questionnaire showed that 22.9% of the respondents used to sell straw directly to buyers, but discontinued this practice because they had no surplus straw, started using straw to fertilise field, or gave another explanation (33.3% each answer). As many as 85.7% of the questioned farmers do not sell straw for energy purposes. The most common reasons are the lack of surplus straw at the farms (48.5%), and the use of ploughing in straw to fertilise soil (33.3%). Few farmers (6.1%) explained that there was no demand for straw, and the price offered was too low (also, 6.1%). Our study proved that selling straw in the district of Braniewo is an unprofitable business due to the lack of market and unattractive prices. Moreover, many farmers need straw for farming purposes, which means that little surplus straw is produced.

Our respondents from the district of Braniewo were also asked to say whether they used other types of renewable energy sources to supply their farms with energy. The answers indicate that 31.4% of the respondents used biomass, e.g., wood, straw, pellets, etc., while 11.4% had solar panels fitted to produce hot water. The smallest share (2.9%) of the respondents used photovoltaic installations, which enabled them to generate electric power.

More than half of the research participants (54.3%) did not give their opinion about the use of renewable energy resources on farms. It would be advisable to consider what causes such poor interest in renewable energy sources in the district of Braniewo. Is it because farmers and entrepreneurs are of the opinion that using such energy resources is unprofitable? Or is it because the capital needed to make the necessary investments exceeds the financial capacities of farmers in the district of Braniewo, which is why they continue to use conventional energy resources?

This study has shown that very few of the respondents from the district of Braniewo cultivate or else cultivated in the past crops for energy purposes, which is why they were asked to make an assessment of the factors that, in their opinion, act as barriers to the production of non-food energy crops (e.g., willow, miscanthus, etc.).

The factors which, according to the respondents, most strongly limited production of non-food energy crops were given in the following order: Low profitability of such plantations (in total 84.0% of the answers, including 56.0% highly and 28.0% moderately affecting), the absence of a system of incentives for farmers to start growing energy plants (e.g., subsidies) (in total 73.1%, including 42.3% strongly agreeing and 30.8% agreeing), and inadequate possibilities to sell plants for energy generation (in total 57.7%, including 30.8% strongly agreeing and 26.9% agreeing). At the same time, the respondents concluded that certain factors did not have a limiting effect; for example, inadequate natural conditions for growing energy crops (60.9%), little knowledge about energy crops and their plantations (54.2%), and bad experiences growing plants for energy purposes (47.8%). The research results demonstrated, not for the first time, that production of energy crops in the district of Braniewo is limited due to the lack of profitability, absence of market to sell energy crops, and the lack of financial support to such plantations.

This survey-based study also revealed the opinion of the respondents about factors which influence the development of feedstocks for the generation of renewable energy. The following were considered to be the most important contributors: The need to reduce pollution (in total 68.0%, including 36.0% citing it as a strong influence and 32.0% a moderate influence), a wish to limit the costs of energy consumption on farms (in total 60%, 23.1% citing it as a strong influence and 30.8% a moderate influence), and a wish to diversify revenues (in total 53.9%, including 23.1% claiming it as a strong influence and 30.8% a moderate influence). The factor which, according to 34.8% of the respondents,

did not have any influence on the development of production of feedstocks for renewable energy generation, consisted of the natural conditions for growing energy crops.

Because production of feedstocks for generation of renewable energy, as well as renewable energy generation itself, can have a positive influence on the sustainable socio-economic development of a region, the respondents were asked to say whether public support would affect the development of this type of production. The results show that 54.3% of the respondents agreed that public support was necessary for further development of the production of feedstocks for generation of renewable energy. Most indicated the support in the form of subsidies (45.4%), while fewer (39.4%) suggested legal regulations pertaining to the production of energy raw materials. Other forms indicated by the respondents were: The creation of a market for selling such raw materials, implementing guarantees of selling such raw materials at profitable prices. On the other hand, only 2.9% of the respondents declared that public support in this area was unnecessary; as many as 28.6% had difficulty answering the question, and 14.3% did not answer it at all.

## 4. Conclusions

Renewable energy sources are a priority direction in the development of energy generation on all the local, regional, and higher levels. Biomass is a typical local fuel, which should be utilised by local individual energy consumers, and which should be the primary fuel for distributed cogeneration plants producing electric power and thermal energy.

The theoretical, potential yield of straw in the district of Braniewo is 83,305 t a year. To estimate the volume of straw which could be used for energy purposes, the amount of straw used in agriculture, namely for bedding, feeds and maintenance of a balanced content of organic substance in soil, was subtracted from the amount of harvested straw.

The calculations verified that there is surplus straw produced in the district of Braniewo, exceeding the agricultural demand for straw. On average, per year, about 41,531 t of straw, having the market value of €1.3 million, remains in the analysed district, and is available for energy generation. The surplus production of straw, after some of its yield has been used for farming, corresponds to an amount of energy ca. 602,202 GJ. Assuming that the calorific value of medium quality coal is 25 MJ·kg$^{-1}$, this corresponds to 24,088 t of coal. Should the average price of coal be 25 MJ·kg$^{-1}$ €·t$^{-1}$, the monetary value of the surplus straw as a fuel substitute equals around 4.3 million euro. By processing the surplus straw into pellets, its value would increase to around 8 million euro (at the price for pellets in the Province of Warmia and Mazury equal approximately 193 €·t$^{-1}$) [70].

The farmers who, in our questionnaire, evaluated the factors influencing the growth of production of raw materials for generating renewable energy in the territory of the district of Braniewo, considered the following to be most significant: The need to reduce pollution (in total 68%, including 36.0% saying it was a strong influence and 32.0% a moderate influence), the intention to lower costs of energy consumption on farm (in total 60%; 32.0 affirmed it as a strong influence and 28.0% a moderate influence), and the wish to diversify revenues (in total 53.9%; 23.1% saying it was strong influence and 30.8% a moderate influence). The factor which over 1/3 of the respondents decided had no influence on the growth in the production of agricultural feedstocks for the renewable energy sector was having suitable natural conditions for growing energy crops.

The respondents noted more barriers than opportunities in the field of production of crops for energy purposes (including low profitability of this type of production—84.0% of responses; limited possibilities of selling plant raw material for energy production—57.7% of answers; and lack of financial support to this type of production).

In the district of Braniewo, there is some potential for setting up and running plantations of energy crops. However, it is not used adequately by farmers. For this direction of agricultural production to become one the priorities in agriculture, it would need to be made more popular among local farmers.

**Author Contributions:** Conceptualization, R.M.-B. and S.B.; data curation, S.B., A.N. and K.R.; formal analysis, A.N. and K.R.; investigation, R.M.-B.; methodology, R.M.-B. and S.B.; project administration, S.B.; supervision, R.M.-B.; writing—original draft, R.M.-B. and S.B.; writing—reviewing and editing, A.N.

**Funding:** Survey research was co-financed by the National Support Centre for Agriculture (the Polish state legal entity that is an executive agency).

**Conflicts of Interest:** The authors declare no conflict of interest. The funders had no role in the design of the study; in the collection, analyses, or interpretation of data; in the writing of the manuscript, or in the decision to publish the results.

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
