# Peer review of "Straw Stocks as a Source of Renewable Energy. A Case Study of a District in Poland"

_sustainability, doi:10.3390/su11174714_

Round 1
Reviewer 1 Report
Overall the paper is interesting and well written.
Before publication, the authors should consider to include some additional information concerning the following issues:
1) What is the assumed price of the straw? This has a significant impact on the economic viability for energy use. Is there some copmeting demand, which could increase the price. e.g. horse stables need significant amounts of straw and can paw much higer prices than other users. The can also afford longer transport chains.
2) What is the assumed diameter of the collection area in km? The transport distance to the power station does not only influence the price but the CO2 balance as well.
Author Response
The authors would like to thank the Reviewer for the effort put into writing review and for the valuable remarks and suggestions which enabled us to improve the original manuscript.
The remarks have been taken into account in preparing the current version of the paper. The average market price of straw in transactions done between agricultural producers, as of second quarter of 2019 in the Province of Warmia and Mazury, was included in our calculations.
We did not find in the district submitted to our analysis any business enterprises which would express the need to use straw for energy purposes (heat generating plants using biomass, or other economic entities e.g. producers of substrates for mushroom growing, or producers of insulation materials).
It is assumed that the profitable distance for straw transport for energy purposes is max. 60 km, because straw is a low energy concentrated raw material. In the article was established that straw is a typical, local fuel and should be utilised at a site of production by individual customers.

Reviewer 2 Report
The manuscript presents a sustainable calculation of the straw available for energy purposes in a Polish district. The work is clear, well written and with a correct use of English. I would recommend its publication as it is after taking into account little annotations:
i would suggest to introduce the problematic of the use of straw for energy recovery purposes, as for example the abrasive character of the feedstock that could damage the valorisator or the most probably necessary pre-treatment. how were all the statistics contained on the tables made? i would suggest to shorten the conclusions so they present only the most remarkable results
Author Response
The authors would like to thank the Reviewer for the effort put into writing review and for the valuable remarks and suggestions which enabled us to improve the original manuscript. The remarks have been taken into account in preparing the current version of the paper.
In Poland, straw for energy purposes is most often used in distributed energy plants, in furnaces where pressed straw is burnt. These furnaces are adopted to the requirements of users. Depending on specific needs, they are designed as having appropriate size and shape, and can be fed straw briquettes or bales. Thus, the pretreatment costs of preparing straw as raw material for combustion are eliminated.
Under the tables we now included the sources of data which we employed in our calculations.
The conclusions are shortened, foregoing less important research results to achieve the assumed research goal.
